# Overcome Double Trouble: Baloxavir Marboxil Suppresses Influenza Thereby Mitigating Secondary Invasive Pulmonary Aspergillosis

**DOI:** 10.3390/jof8010001

**Published:** 2021-12-21

**Authors:** Laura Seldeslachts, Cato Jacobs, Birger Tielemans, Eliane Vanhoffelen, Lauren Van der Sloten, Stephanie Humblet-Baron, Lieve Naesens, Katrien Lagrou, Erik Verbeken, Joost Wauters, Greetje Vande Velde

**Affiliations:** 1Biomedical MRI Unit/MoSAIC, Department of Imaging and Pathology, KU Leuven, 3000 Leuven, Belgium; laura.seldeslachts@kuleuven.be (L.S.); birger.tielemans@kuleuven.be (B.T.); eliane.vanhoffelen@kuleuven.be (E.V.); 2Laboratory for Clinical Infectious and Inflammatory Disorders, Department of Microbiology, Immunology and Transplantation, KU Leuven, 3000 Leuven, Belgium; cato.1.jacobs@uzleuven.be (C.J.); lauren.vandersloten@uzleuven.be (L.V.d.S.); Joost.wauters@uzleuven.be (J.W.); 3Laboratory of Adaptive Immunity, Department of Microbiology, Immunology and Transplantation, KU Leuven, 3000 Leuven, Belgium; stephanie.humbletbaron@kuleuven.be; 4Laboratory of Virology and Chemotherapy, Rega Institute, Department of Microbiology, Immunology and Transplantation, KU Leuven, 3000 Leuven, Belgium; lieve.naesens@kuleuven.be; 5Laboratory of Clinical Bacteriology and Mycology, Department of Microbiology, Immunology and Transplantation, KU Leuven, 3000 Leuven, Belgium; katrien.lagrou@uzleuven.be; 6Department of Imaging and Pathology, KU Leuven, 3000 Leuven, Belgium; erik.verbeken@kuleuven.be

**Keywords:** influenza-associated pulmonary aspergillosis (IAPA), baloxavir marboxil, multimodal preclinical imaging-supported mouse model, *Aspergillus fumigatus*, influenza

## Abstract

Influenza-associated pulmonary aspergillosis (IAPA) is a global recognized superinfection in critically ill influenza patients. Baloxavir marboxil, a cap-dependent endonuclease inhibitor, is a newly approved anti-influenza therapeutic. Although the benefits as a treatment for influenza are clear, its efficacy against an influenza-*A. fumigatus* co-infection has yet to be determined. We investigated the therapeutic effect of baloxavir marboxil in a murine model for IAPA. Immunocompetent mice received intranasal instillation of influenza A followed by orotracheal inoculation with *Aspergillus fumigatus* 4 days later. Administration of baloxavir marboxil or sham was started at day 0, day 2 or day 4. Mice were monitored daily for overall health status, lung pathology with micro-computed tomography (µCT) and fungal burden with bioluminescence imaging (BLI). In vivo imaging was supplemented with virological, mycological and biochemical endpoint investigations. We observed an improved body weight, survival and viral clearance in baloxavir marboxil treated mice. µCT showed less pulmonary lesions and bronchial dilation after influenza and after *Aspergillus* co-infection in a treatment-dependent pattern. Furthermore, baloxavir marboxil was associated with effective inhibition of fungal invasion. Hence, our results provide evidence that baloxavir marboxil mitigates severe influenza thereby decreasing the susceptibility to a lethal invasive *Aspergillus* superinfection.

## 1. Introduction

Over the past four decades, viral associated pulmonary aspergillosis (VAPA), such as influenza-associated pulmonary aspergillosis (IAPA) or more recently COVID-19-associated pulmonary aspergillosis (CAPA), have become new entities in the world as serious life-threatening infectious diseases [1,2,3,4,5,6,7,8,9,10]. The lethal combination of influenza and *Aspergillus* leads to a poor outcome characterized by a longer ICU stay and an almost doubled mortality rate in comparison with severe influenza patients [4,5,11]. Understanding the synergism between the virus and the fungus is of utmost importance. We recently developed a preclinical animal model of IAPA disease, confirming influenza as a strong risk factor for developing invasive pulmonary aspergillosis (IPA) in immunocompetent mice [12,13].

Ever since the flu pandemic of 2009, the neuraminidase inhibitor (NAI) oseltamivir has been used as standard-of-care treatment to suppress influenza. This drug prevents the release and spread of the influenza virus from the host cell by inhibiting the viral neuraminidase. We found that early oseltamivir modulated the severity of influenza thereby mitigating the susceptibility to IAPA in an immunocompetent murine model [12]. In addition, in the clinical setting, prophylactic oseltamivir proved to be effective in prevention of IAPA in children after haploidentical hematopoietic cell transplantation [14]. However, despite the beneficial effects of oseltamivir, we need to have alternative antivirals at hand. Indeed, it is hypothesized that oseltamivir may play a harmful role in the host immune defense against fungal superinfection. Human and mice express neuraminidases (NEU1-NEU4), which show similar tertiary structures and active-site amino acids as the viral sialidases. NAIs like oseltamivir, might therefore inhibit the endogenous neuraminidase activity on the host immune cells and modulate host-pathogen interaction [15,16]. Although only limited inhibitory effects of oseltamivir on mammalian sialidases were reported [15], Dewi et al. (2021) showed that oseltamivir influences the host response against *A. fumigatus* in vitro by inhibiting the endogenous neuraminidases resulting in an impaired capacity of human peripheral blood mononuclear cells (PBMCs) and murine splenocytes to kill *Aspergillus* [16]. Furthermore, in vivo, an increased susceptibility to IPA with higher fungal burden and mortality was found [16].

In addition to a possible effect of blocking the host neuraminidase by NAI, emergence of NAI resistant influenza strains can occur that warrant the need for alternative antivirals. Currently, resistance to NAI in community isolates remains at low levels (<1%), however higher incidences of oseltamivir-resistant variants exist [17,18]. In 2009, the influenza A (H1N1)pdm09 virus with the NA-H275Y amino acid substitution occurred as a widespread oseltamivir-resistant influenza strain [17,19,20,21]. In a French multicenter observational cohort study, a highly reduced sensitivity to oseltamivir was found in 23% of the patients with a severe influenza A(H1N1)pdm09 [NA-H275Y] pneumonia [19]. If resistance occurs, it will be important to use other effective antiviral treatments which can lower the severity of the influenza infection and thereby mitigate the susceptibility to IPA [12]. Resistance together with the possible effect of blocking the host neuraminidase indicate the importance to preclinically assess new antiviral drugs, relying on different modes of actions, on the effectiveness in IAPA.

Baloxavir marboxil, a cap-dependent endonuclease inhibitor, is a novel anti-influenza therapeutic approved by Japan and the U.S. Food and Drug Administration (FDA) in 2018 and the European Medicine Agency (EMA) in 2021 [22,23]. Different clinical trials have shown the superior effect of a single-dose baloxavir marboxil in alleviating influenza symptoms compared to placebo and reducing viral load compared to oseltamivir and placebo [24,25]. Additionally in preclinical studies, oral administration of baloxavir marboxil (15 mg/kg, twice daily) was more efficient than oseltamivir in reducing the mortality rate, body weight loss and viral titers, even with a delay in treatment initiation of 4 or 5 days post-influenza infection in mice [26,27]. Although the benefits of baloxavir marboxil as a treatment method for influenza are well described, the effectiveness of the antiviral drug against an influenza-*A. fumigatus* co-infection has yet to be determined.

Hence, we investigated the time-dependent treatment effect of baloxavir marboxil in suppressing influenza and its efficacy to prevent a secondary invasive pulmonary aspergillosis in an imaging-based immunocompetent murine model of IAPA.

## 2. Materials and Methods

### 2.1. Influenza Virus and Aspergillus Strain

A mouse-adapted influenza virus A/H3N2/Ishikawa/7/82 and a red-shifted thermostable firefly luciferase *A. fumigatus* strain were prepared, as previously described [12] (manuscript in consideration [28]).

### 2.2. Mouse Model

All animal experiments were approved by the KU Leuven Ethical Committee for animal research (license p074/2018). Mice were kept in individually ventilated cages with free access to food and water in a conventional animal facility. An antibiotic (50 mg/kg/day, Baytril^®^, Bayer, Leverkusen, Germany) was added to the drinking water to prevent bacterial infections. Eight- to ten-week-old immunocompetent male C57BL/6NTac mice (internal stock of the Laboratory Animal Centre, KU Leuven, Leuven, Belgium) were randomly assigned to experimental groups with a sample size of 5 per group per experiment. The experiment was repeated twice. The mouse model was developed as previously described [12]. Briefly, mice received intranasal instillation of 10 plaque forming units (PFU) influenza virus on day 0 (10 µL per nostril) and orotracheally inoculation with 20 µL of *A. fumigatus* spore solution (5 × 10^9^ conidia/mL) under inhalation anesthesia with isoflurane (1.5–2% in 100% oxygen, Primal Critical Care) on day 4. Baloxavir marboxil (15 mg/kg) or phosphate-buffered saline (PBS) treatment was administered twice daily by oral gavage (Figure 1A). The dose of 15 mg/kg twice daily mimic the plasma concentration of 6.85 ng/mL in humans and therefore used to predict clinical effectiveness [26,27]. During the experiments, mice were longitudinally monitored for body weight and non-invasively scanned with micro-computed tomography (*n* = 10 per group) and bioluminescence imaging (*n* = 5 per group). Mice were anesthetized by inhalation of 1.5–2% isoflurane in 100% oxygen during scanning. Sacrifice was performed at a predefined endpoint on day 7, with an overdose of pentobarbital via intraperitoneal injection. Bronchoalveolar lavage (BAL) fluid, right and left lung lobes were collected as previously described [12]. Left and right lungs were used for respectively histopathology (*n* = 5 per group) and quantification of virus titers and fungal burden (*n* = 10 per group).

### 2.3. Micro-Computed Tomography (µCT)

Acquisition, visualization and quantification of µCT were performed as previously described [12]. Briefly, for µCT, a SkyScan 1278 (Bruker µCT, Kontich, Belgium) was used for acquisition with the following scan parameters (50 kVp X-ray source, 1 mm aluminum X-ray filter, 350 µA current, 150 ms exposure time per projection, 0.9° increments over a total angle of 220°). Visualization of reconstructed µCT were performed with DataViewer, CTan and CTvox. Semi-quantitative scoring was performed by two blinded researchers according to Seldeslachts et al. (2021) [12]. For quantification, imaging-derived biomarkers of lung pathology (non-aerated lung volume (NALV)) was determined based on manually delineation of region of interest (ROI) resulting in a volume of interest (VOI) covering the entire lung and the previously determined threshold of 108–255 greyscale. Bronchial dilation was determined by manually delineation of ROI of the airways and a threshold of 0–90 greyscale.

### 2.4. Bioluminescence Imaging (BLI)

For in vivo and ex vivo BLI, data acquisition, visualization and quantification were performed as previously described [12]. Briefly, the IVIS Spectrum System (Perkin-Elmer, Hopkinton, MA, USA) was used with following settings (exposure time 1 min, medium binning, F/stop of 1 and subject height of 1.5 cm in vivo and 1 cm for ex vivo). Quantification of BLI data was performed with Living Image Software version 4.7.3 as previously described [12]. The highest total photon flux from consecutive acquisitions was used for further analysis.

### 2.5. Quantification of Virus and Fungal Titers

Viral and fungal load detection was similar as Seldeslachts et al. (2021) [12]. Briefly, for fungal load detection by colony-forming unit (CFU) counting, serial dilutions of right lung homogenates were plated on Sabouraud agar plates and incubated at 37 °C for 48 h. Reported CFU counts represent average values of duplicate plating of each dilution. Viral titer determination was performed by adding a serial dilutions of right lung homogenate supernatants to Madin–Darby Canine Kidney cells (MDCK) in quadruplicate, incubation at 35 °C during 72 h and assessment of a cytopathic effect by microscopy scoring.

### 2.6. Histopathology

Each left lung lobe (*n* = 5 per group) was fixed with 10%-formalin, 24 h post fixated and then stored in PBS-0.1% sodium-azide at 4 °C. Subsequently, samples were paraffin embedded and 5 µM sagittal sections were stained with hematoxylin-eosin (H&E) and Grocott’s methenamine silver staining (GMS). Histological tissue sections were scored by a blinded pathologist (EVB).

### 2.7. Statistical Analysis

Statistics were performed using GraphPad Prism (version 8.1.2, GraphPad Software, San Diego, CA, USA). Longitudinal data were analyzed with two-way ANOVA with Tukey’s multiple comparison test (repeated measurements performed from day 0–day 4 and day 4–day 7) and endpoint data with one-way ANOVA with Tukey’s multiple comparisons test or Kruskal–Wallis test. All graphs represent mean values ± SD. Differences were considered significant if *p* was smaller or equal to 0.05. * *p* < 0.05, ** *p* < 0.01, *** *p* < 0.001, **** *p* < 0.0001. *n* values represent the number of animals.

## 3. Results

### 3.1. Baloxavir Marboxil Treatment Improves Survival and Limits Body Weight Loss

Recently, we identified influenza as a strong risk factor for developing invasive pulmonary aspergillosis (IPA) in immunocompetent mice [12]. To assess the time-dependent treatment effect of baloxavir marboxil on the severity of influenza and subsequent susceptibility to IPA in this double-hit mouse model, we treated mice with baloxavir marboxil or PBS started at day 0, day 2 or day 4 up to day 7 (Figure 1A). During the influenza infection (i.e., day 0–day 4), we observed markedly less body weight loss in the baloxavir marboxil prophylactic treated group in comparison with other groups (Figure 1B). Upon *Aspergillus* co-infection (i.e., day 4–day 7), the non-treated mice showed a faster and more substantial body weight loss (Figure 1B) as well as a fatal clinical deterioration in 70% of the animals by day 7, the predetermined endpoint of this study (Figure 1C). In sharp contrast, all baloxavir marboxil-treated mice experienced only a moderate body weight loss (Figure 1B) and less animals reached the humane endpoint at day 7 (Figure 1C) the earlier treatment was started. Hence, baloxavir marboxil limits body weight loss and improves survival in a treatment-onset dependent manner.

### 3.2. Baloxavir Marboxil Reduces Infiltrates and Bronchial Dilation on Lung µCT after Influenza and Aspergillus Co-Infection

To further investigate the impact of baloxavir marboxil on the lung lesion development during influenza pneumonia and upon *Aspergillus* superinfection, we longitudinally followed up the mice through non-invasive µCT. Viral induced lung pathology and bronchial dilation (day 0–day 4) were significantly reduced in prophylactic treated mice in comparison with other groups, coinciding with only a moderate body weight loss (Figure 2A,B,E). During *Aspergillus* co-infection (day 4–day 7), visual assessment of 3D rendered aerated lung volumes showed severe diffuse pulmonary consolidations of entire lung lobes and bronchial dilation in non-treated mice while baloxavir marboxil treatment resulted in less to no pulmonary infiltrates or bronchial dilation the earlier treatment was started (Figure 2A,B,E). In line with the visual observations, the quantification of longitudinal µCT-derived biomarkers (non-aerated lung volume (NALV) and bronchial dilation) confirmed our findings of a time-dependent reduction in viral and viral-fungal pulmonary lesion development, as well as bronchial dilation during baloxavir marboxil treatment (Figure 2C,D,F). Together, these results show that baloxavir marboxil limits lung pathology on µCT after influenza and after *Aspergillus* co-infection.

### 3.3. Baloxavir Marboxil Effectively Clears Influenza

Next, we determined the viral load in lung tissue at endpoint in treated and non-treated mice (Figure 3). The viral load at day 7 was high in untreated group, while in the baloxavir marboxil treated groups the viral load was significantly reduced regardless of the of the timing of baloxavir marboxil administration. So baloxavir marboxil has an effective viricidal activity against influenza.

### 3.4. Baloxavir Marboxil Reduces the Risk for a Secondary Invasive Pulmonary Aspergillosis

Having established fewer lung pathology and viral clearance after baloxavir marboxil treatment, we next explored the effectiveness in inhibiting invasive fungal disease development through longitudinal BLI of luciferase-tagged *A. fumigatus* as a proxy of fungal burden and CFU counts. In vivo pulmonary bioluminescence signal intensity was significantly reduced in all baloxavir marboxil treated mice compared with non-treated mice towards day 7 (Figure 4A,B). Furthermore, the fungal load in lung homogenates of prophylactic treated mice was significantly lower than in non-treated mice, measured by ex vivo BLI (close to background signal) and CFU (Figure 4C,D). Strikingly, *A. fumigatus* was detected in all groups. Histopathology confirmed the presence of *A. fumigatus* conidia in all groups but identified a clear morphological/phenotypical difference (Figure 4E). In baloxavir marboxil treated mice no airway invasive hyphae were detected while clearly present in non-treated mice (Figure 4E). These data suggest that baloxavir marboxil is effective in preventing a secondary invasive pulmonary aspergillosis.

### 3.5. Baloxavir Marboxil Treatment Onset-Dependent Reduction in Lung Pathology and Local Immune Response

We further zoomed in on the effectiveness of baloxavir marboxil in preventing severe lung pathology and extensive local immune response after influenza-*Aspergillus* co-infection. We found a clear morphological difference between treated and non-treated mice characterized by a treatment-onset dependent drastic decrease in distribution of affected lung tissue and less severe to no epithelial damage (erosion versus stimulation/cytotoxicity) (Figure 5A,B). Histological general inflammatory subset analysis of mono- versus polymorphonuclear leukocyte populations showed a marked decrease in neutrophils in treated mice compared to non-treated mice (Figure 5C,D). The same observation was found in bronchoalveolar lavage fluid (BALF) where baloxavir marboxil treatment resulted in a treatment-onset dependent decrease in neutrophil infiltration. In BALF this decrease in neutrophil infiltration was in favor of the mononuclear infiltrate macrophage population (Figure 6A,B). Collectively, these results show less severe epithelial damage, limited pneumonic extension and a reduced local immune response in baloxavir marboxil treated mice with a treatment-onset dependent pattern.

## 4. Discussion

Here we report the therapeutic effectiveness of baloxavir marboxil in suppressing influenza and thereby preventing a secondary invasive pulmonary aspergillosis in a non-immunocompromised host. We herewith corroborate influenza as an independent risk factor for developing IPA.

The antiviral drug, baloxavir marboxil, is known to show therapeutic activity in uncomplicated influenza patients and preclinical models of influenza A and B virus infections, including resistant strains [24]. On top of this, we now preclinically show that baloxavir marboxil is effective during an influenza-*A. fumigatus* superinfection. This is evidenced by firstly an improved survival, ameliorated body weight and suppressed replication of influenza virus in the lungs even when treatment with baloxavir marboxil was delayed. Several preclinical and clinical studies in a mono-infection setting have already shown a time-dependent reduction in lung viral titers, decreased mortality and improved body weight after baloxavir marboxil treatment [24,25,26,27]. We now demonstrated that this is also the case after fungal co-infection in a preclinical mouse model of IAPA. Secondly, we longitudinally visualized for the first time a baloxavir marboxil treatment-onset dependent reduction in pulmonary lesions and bronchial dilation after influenza and after *Aspergillus* co-infection. Thirdly, limited influenza severity in mice treated with baloxavir marboxil was associated with reduced fungal burden development and morphological absence of airway invasive hyphae in the lungs. Finally, antiviral therapy reduced the excess infiltration of inflammatory neutrophils into the lung, thus protecting the lungs from severe pneumonic damage as well as an overreacting local immune response in bronchoalveolar lavage fluid. Additionally, baloxavir marboxil treatment was associated with higher levels of alveolar macrophages in BALF. This could point to a better anti-fungal lung macrophage defense and thus lower fungal burden [29,30]. The significant lower number of alveolar macrophages in non-treated mice could point to a potential immune dysregulation through which influenza predisposes to a fungal superinfection. It is known that type I IFN induced by influenza can inhibit the production of the chemokine CCL2, which is important for macrophage recruitment [31,32]. Besides an effect on recruitment, influenza virus infection can result in depletion of alveolar macrophages and can impair the macrophage phagocytic activity important for fungal clearance [31,32,33]. However, further comprehensive immunological studies are needed to identify the immunological mechanism behind IAPA. Thus, baloxavir marboxil is effective in decreasing susceptibility to influenza-associated pulmonary aspergillosis in a murine model even when treatment was delayed. We herewith preclinically show that baloxavir marboxil forms a good alternative antiviral drug which can be used in case oseltamivir resistant strains occur and we re-validate the observation of Schauwvlieghe et al. (2018) in which severe influenza was identified as an independent risk factor for IPA-development in immunocompetent hosts.

Besides forming a good alternative antiviral drug, baloxavir marboxil, also has a different mechanism of action. As an influenza cap-dependent endonuclease inhibitor, baloxavir marboxil bypasses the possible negative effect related to the use of oseltamivir. Indeed, it is hypothesized that the neuraminidase inhibitor oseltamivir might inhibit the endogenous neuraminidase activity on the host immune cells although only limited inhibitory effects of oseltamivir on mammalian sialidases were reported [15,16,34]. Dewi et al. (2021) recently reported a detrimental effect of oseltamivir on fungal killing capacity of mouse splenocytes and human PBMCs in vitro [16]. Furthermore, in vivo, an increased susceptibility to IPA with higher fungal burden and mortality was found in oseltamivir treated mice [16]. However, although oseltamivir is suspected to have a potential adverse effect, our preclinical studies of oseltamivir and baloxavir marboxil, demonstrate the positive effect on influenza disease symptoms which counteracts the potential adverse effect on the fungal host response and the risk of acquiring IAPA [12].

This study highlights the importance of the timing of the baloxavir marboxil treatment. The earlier the treatment is started the better. However, in clinical practice, this is not easy to realize. The majority of the critically ill influenza patients in ICU have had influenza symptoms for more than a few days [35,36]. We now preclinically show that even if baloxavir marboxil treatment is delayed until 4 days post infection, we still observe marked improvements in our IAPA mice.

## 5. Conclusions

In conclusion, our results provide evidence that initiating early baloxavir marboxil can prevent a severe influenza infection thereby decreasing the susceptibility to a lethal *Aspergillus* co-infection. In the future, clinical studies can explore the therapeutic potential of baloxavir marboxil in fighting IAPA even when treatment-onset is postponed as early treatment is not always possible in clinical situation. In the broader context, we emphasize the importance to preclinically investigate potential pathogenic mechanisms and therapeutic interventions not only in IAPA but also in other VAPA’s.

## Figures and Tables

**Figure 1 jof-08-00001-f001:**
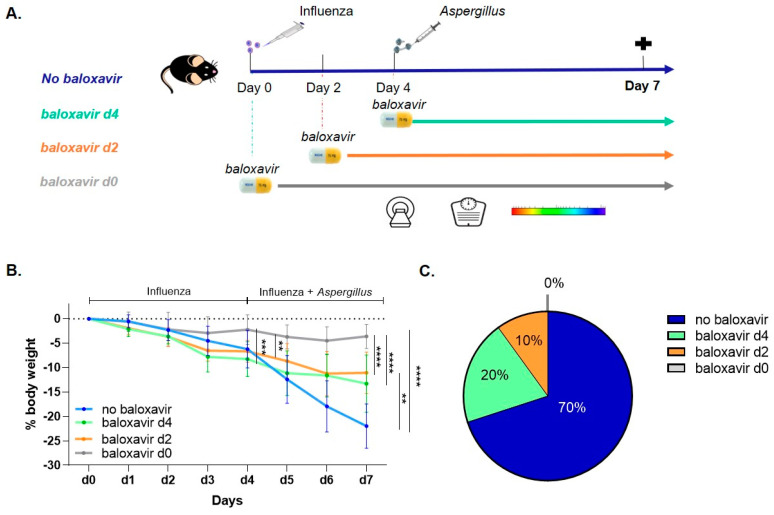
Baloxavir marboxil treatment improves survival and body weight loss. (**A**) Experimental setup: immunocompetent mice received intranasal instillation influenza A followed by orotracheal inoculation with *Aspergillus fumigatus* 4 days later. Baloxavir marboxil treatment or sham (no baloxavir, blue) was started at day 4 (green), day 2 (orange) or day 0 (grey) up to day 7. Mice were longitudinally monitored for body weight, with micro-computed tomography (µCT) and bioluminescence imaging (BLI). (**B**) Relative body weight evolution of influenza/*Aspergillus* co-infected mice receiving sham treatment (no baloxavir, blue, *n* = 10 mice), baloxavir marboxil treatment starting from day 4 (green, *n* = 10 mice), baloxavir marboxil treatment started on day 2 (orange, *n* = 10 mice) and baloxavir marboxil treatment from day 0 (grey, *n* = 10 mice). (**C**) Pie chart representing survival expressed as the percentage (%) of animals reaching humane endpoint (>20% body weight loss) at day 7. Differences were considered significant if *p* was smaller or equal to 0.05. ** *p* < 0.01, *** *p* < 0.001, **** *p* < 0.0001. Data are represented as mean ± SD.

**Figure 2 jof-08-00001-f002:**
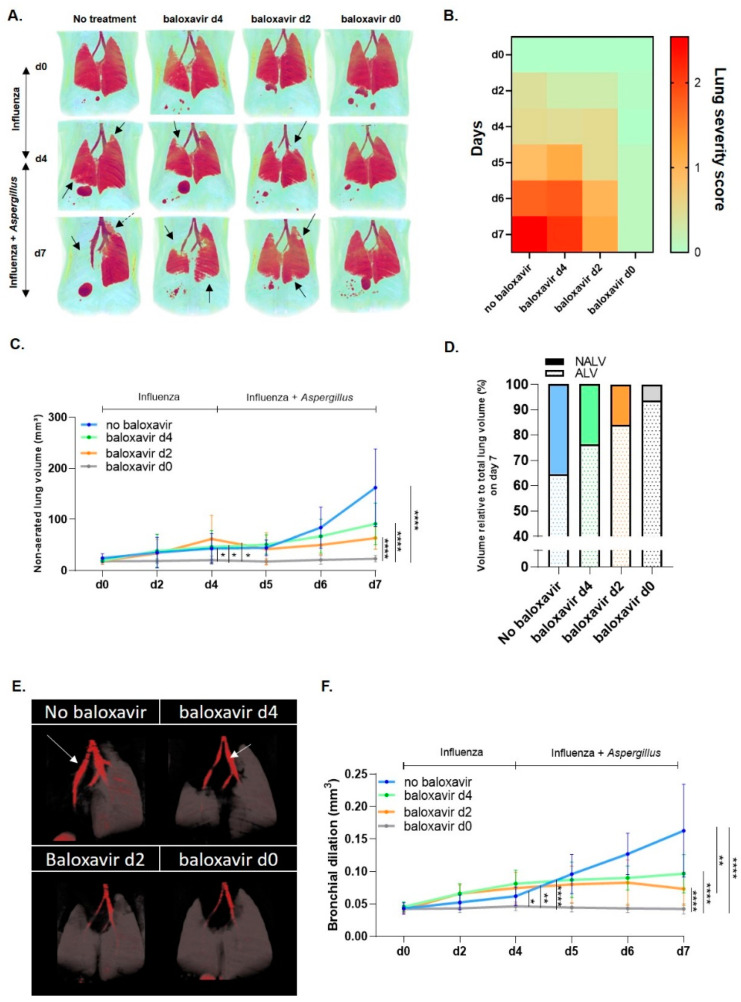
Baloxavir marboxil treatment reduces viral and viral-fungal pulmonary pathology and bronchial dilation. (**A**) Longitudinal three-dimensional (3D) visualization of aerated lung volume acquired with µCT. Red: aerated lung tissue, Black arrows point at pulmonary infiltrates/lung pathology and dashed arrow to bronchial dilation. (**B**) Heatmap representation of lung severity scores based on transverse lungs µCT images of non-treated (sham, no baloxavir), baloxavir marboxil from day 4, baloxavir marboxil from day 2 and baloxavir marboxil from day 0 treated mice. Red: severe pulmonary lesions, orange: moderate pulmonary lesions, green: no pulmonary lesions. (**C**) Graph represents longitudinal µCT-derived biomarker “non-aerated lung volume” (pulmonary lesions) of immunocompetent influenza/*Aspergillus* co-infected mice that received no baloxavir (sham, blue, *n* = 10 mice), baloxavir marboxil treatment-onset from day 4 (green, *n* = 10 mice), baloxavir marboxil treatment-onset from day 2 (orange, *n* = 10 mice) and baloxavir marboxil treatment-onset from day 0 (grey, *n* = 10 mice). (**D**) µCT-derived biomarkers: non-aerated lung volume and aerated lung volume relative to total lung volume on day 7 of non-treated (sham, no baloxavir, blue, *n* = 10 mice), baloxavir marboxil from day 4 (green, *n* = 10 mice), baloxavir marboxil from day 2 (orange, *n* = 10 mice) and baloxavir marboxil from day 0 (grey, *n* = 10 mice). (**E**) Longitudinal 3D visualization of aerated lung volume acquired with µCT. Grey: aerated lung tissue, Red: trachea with arrow bronchial dilation. (**F**) Quantification of bronchial dilation determined by manually delineation of the region of interest (ROI) of the airways of mice receiving no baloxavir marboxil (sham, blue, *n* = 10 mice), baloxavir marboxil treatment-onset from day 4 (green, *n* = 10 mice), baloxavir marboxil treatment-onset from day 2 (orange, *n* = 10 mice) and baloxavir marboxil treatment-onset from day 0 (grey, *n* = 10 mice). Statistical analysis for longitudinal graph: two-way ANOVA with Tukey’s multiple comparisons test repeated measurements performed from day 0 until day 4 (influenza infection) and day 4 until day 7 (influenza-*Aspergillus* co-infection). Differences were considered significant if *p* was smaller or equal to 0.05. * *p* < 0.05, ** *p* < 0.01 **** *p* < 0.0001. *n* values represent the number of animals. Data are represented as mean ± SD.

**Figure 3 jof-08-00001-f003:**
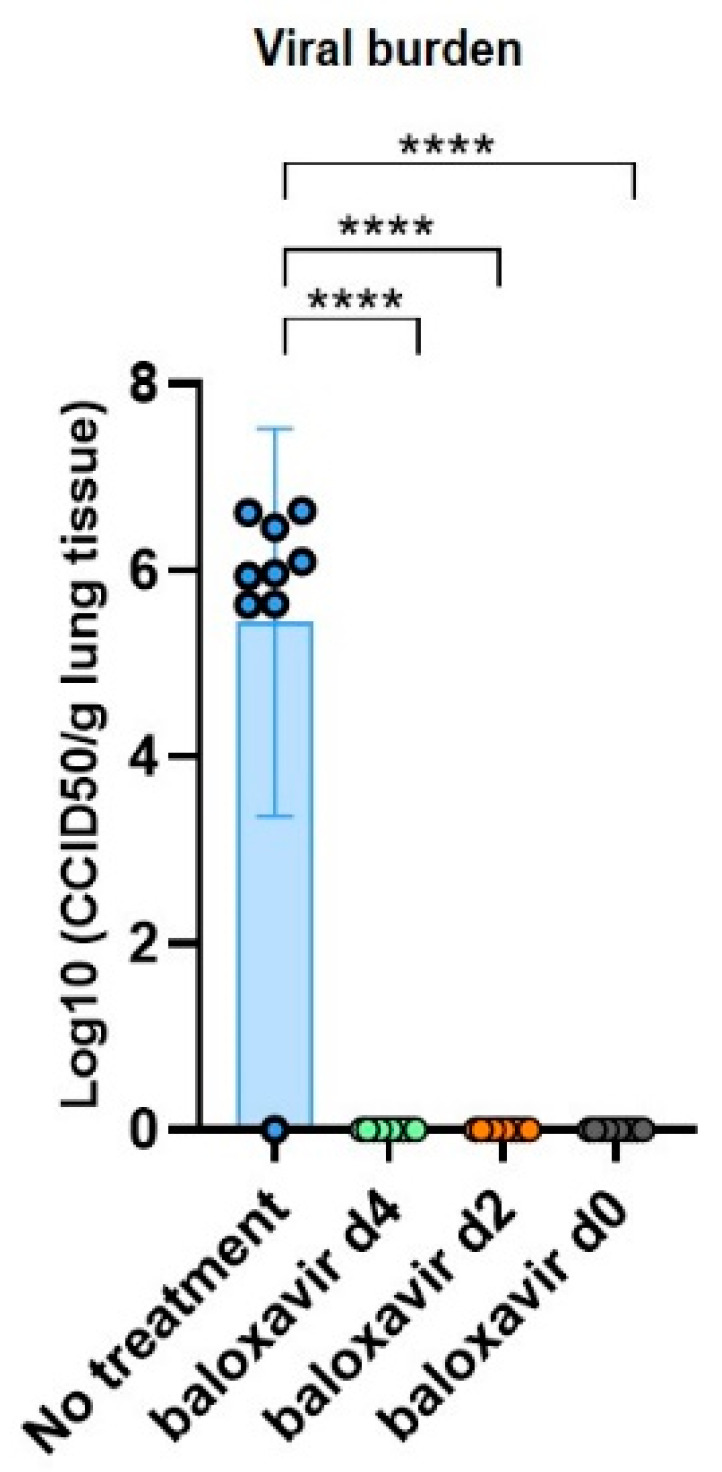
Baloxavir marboxil has an effective viricidal activity against influenza. Viral burden: graph representing the log_10_ CCID50 per gram lung tissue receiving no baloxavir marboxil (sham, blue, *n* = 10 mice), baloxavir marboxil treatment-onset from day 4 (green, *n* = 10 mice), baloxavir treatment-onset from day 2 (orange, *n* = 10 mice) and baloxavir marboxil treatment-onset from day 0 (grey, *n* = 10 mice). Differences were considered significant if *p* was smaller or equal to 0.05. **** *p* < 0.0001. Data are represented as individual data-points, mean ± SD.

**Figure 4 jof-08-00001-f004:**
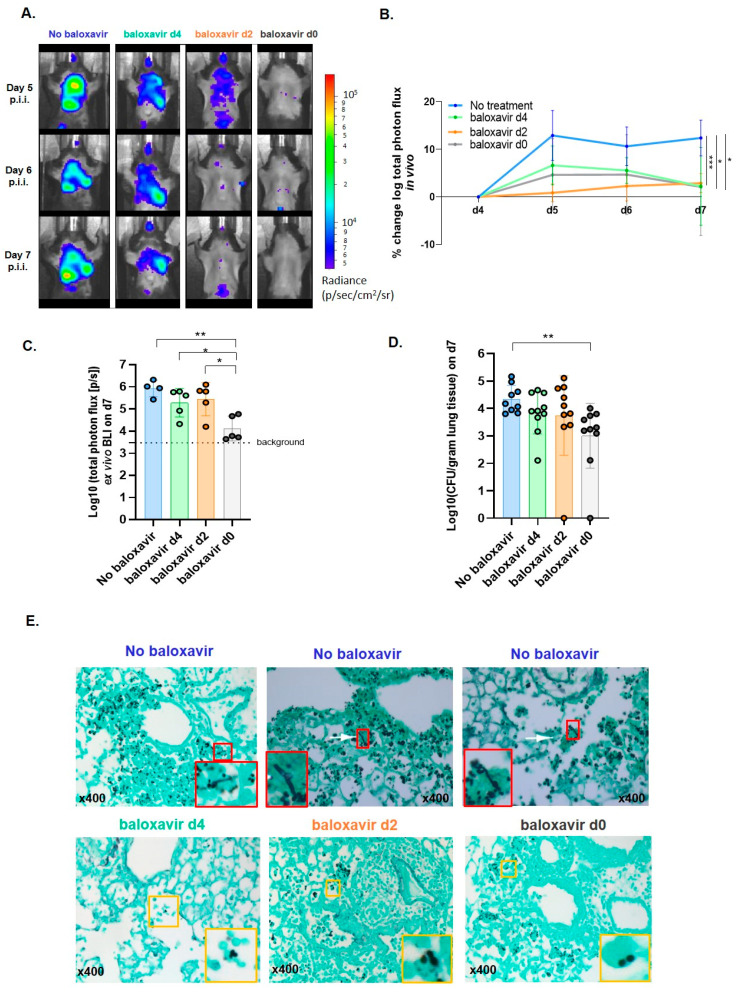
Baloxavir marboxil is effective in preventing secondary invasive pulmonary aspergillosis. (**A**) Bioluminescence images (BLI) of one mouse per group acquired on day 5, day 6 and day 7 post influenza-infection. (**B**) Quantification of total photon flux from in vivo BLI based on ROI covering the complete lung region and represented as the % change of the log_10_ (total photon flux) values in comparison with baseline of influenza-*Aspergillus* day 4 co-infected mice without baloxavir marboxil (blue, *n* = 5 mice), baloxavir marboxil from day 4 (green, *n* = 5 mice), baloxavir marboxil from day 2 (orange, *n* = 5 mice) and baloxavir marboxil from day 0 (grey, *n* = 5 mice). Longitudinal statistical analysis: repeated measurements two-way ANOVA with Tukey’s multiple comparisons test performed from day 4 until day 7. (**C**) Quantification of the total photon flux from ex vivo BL images based on ROI covering the well containing undiluted lung homogenates on day 7 from no baloxavir marboxil (sham, blue, *n* = 5 mice), baloxavir marboxil from day 4 (green, *n* = 5 mice), baloxavir marboxil from day 2 (orange, *n* = 5 mice) and baloxavir marboxil from day 0 (grey, *n* = 5 mice) treated mice. (**D**) Graph representing the log_10_ CFU count per gram lung tissue on day 7 from non-treated (no baloxavir, blue, *n*= 10 mice), baloxavir marboxil from day 4 (green, *n*= 10 mice), baloxavir marboxil from day 2 (orange, *n* = 10 mice) and baloxavir marboxil from day 0 (grey, *n* = 10 mice). (**E**) Histological Grocott’s methenamine silver staining (GMS) (magnification ×400) of left lung of different groups (no baloxavir, baloxavir marboxil treatment from day 4, baloxavir marboxil treatment from day 2, baloxavir marboxil treatment from day 0) sacrificed on day 7. Hyphae (red box), intracellular conidia (yellow box). Differences were considered significant if *p* was smaller or equal to 0.05. * *p* < 0.05, ** *p* < 0.01, *** *p* < 0.001. Data are represented as mean ± SD.

**Figure 5 jof-08-00001-f005:**
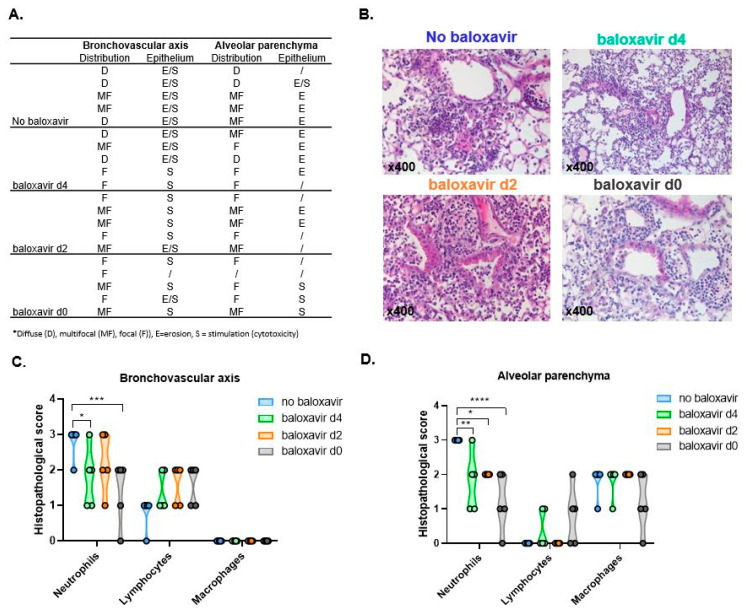
Baloxavir marboxil treatment onset-dependent reduction in lung pathology (**A**) Histopathological scores including the type of distribution (diffuse (D), multifocal (MF), focal (F)) for *n* = 5 mice per group. (**B**) Histological hematoxylin and eosin (H&E) staining (magnification ×400) of no baloxavir marboxil, baloxavir marboxil from day 4, baloxavir marboxil from day 2 and baloxavir marboxil from day 0. (**C**,**D**) Histological general inflammatory subset analysis (neutrophils, macrophages, lymphocytes) in bronchovascular axis (**C**) and alveolar parenchyma (**D**) from non-treated (no baloxavir, blue, *n* = 5 mice), baloxavir marboxil from day 4 (green, *n* = 5 mice), baloxavir marboxil from day 2 (orange, *n* = 5 mice) and baloxavir marboxil from d0 (grey, *n* = 5 mice). Differences were considered significant if *p* was smaller or equal to 0.05. * *p* < 0.05, ** *p* < 0.01, *** *p* < 0.001, **** *p* < 0.0001. Data are represented as mean ± SD.

**Figure 6 jof-08-00001-f006:**
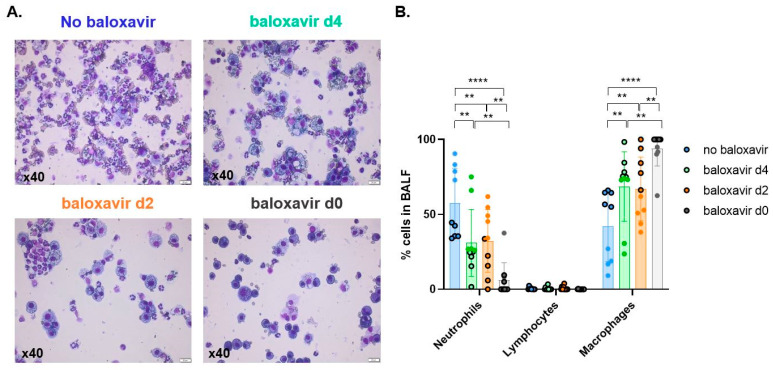
Treatment onset-dependent reduction in local immune response. (**A**,**B**) Visualization (**A**) and quantification (**B**) of mono- and polymorphonuclear leukocyte populations in bronchoalveolar lavage fluid from non-treated (no baloxavir, blue, *n* = 10 mice), baloxavir marboxil from day 4 (green, *n* = 10 mice), baloxavir marboxil from day 2 (orange, *n* = 10 mice) and baloxavir marboxil from day 0 (grey, *n* = 10 mice). Color fill vs. black contour color filled dot represents results from two different experiments. ** *p* < 0.01, **** *p* < 0.0001. Graphs are represented individual data points, mean ± SD.

## Data Availability

Data available upon reasonable request to the corresponding author.

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
