# Peer review of "Overcome Double Trouble: Baloxavir Marboxil Suppresses Influenza Thereby Mitigating Secondary Invasive Pulmonary Aspergillosis"

_jof, 2021, doi:10.3390/jof8010001_

Round 1
Reviewer 1 Report
The manuscript described that a novel influenza anti-influenza therapeutic drug, Baloxavir marboxil not only can prevent a severe influenza infection but also decrease the susceptibility to a lethal invasive Aspergillus co-infection and was well written.
In Section 3.3, it is described that the viral load at day 7 was undetectable in baloxavir marboxil treated mice. Please state that it significantly reduces the viral load regardless of the timing of baloxavir marboxil administration.
How important is the "early reduction" of influenza virus by baloxavir marboxil to prevent secondary invasive pulmonary aspergillosis? Because the early reduction of influenza virus is one of features of effect of baloxavir marboxil.
Reviewer 2 Report
In this study, Seldeslachts and colleagues addressed the role of baloxavir marboxil in the development of experimental influenza-associated pulmonary aspergillosis (IAPA). By using a two-hit mouse model, the authors report the time-dependent therapeutic efficacy of baloxavir against IAPA. This is an interesting and very well-designed proof-of-concept study on a clinically relevant and timely topic and that illustrates the potential of an antiviral with a different mode of action from neuraminidase inhibitors such as oseltamivir, in the context of IAPA. The manuscript is very-well written, and the results are presented in a logical and straightforward manner. I have just a few minor comments:
1- Fig.1 C is an uncommon representation of mice survival after an infection. Is it possible to replace this information with survival curves (similar to the weight loss) so that there is information not only on the mortality rates at day 7, but also on its evolution during the two-hit model?
2- Baloxavir appears to confer protection from IAPA by restraining the preceding viral infection, but to definitely confirm this, one would have to show that baloxavir does not protect from fungal infection in the absence of viral infection. Should the authors have these data available, it would further support their conclusions.
3- The discussion could also include further comments on the potential mechanisms through which the influenza infection predisposes to fungal superinfection, particularly with regard to the immunological data presented by the authors.
Reviewer 3 Report
I read with great interest the paper. First all congratulations to the authors. The paper is well wrote and with good idea research. I appreciate a lot .
Only minor suggestions
- Introduction: well introduce the oseltamivir ant its possible role.
- Methods and result: very well wrote. Figure are excellent
- Discussion: in COVID pandemic period could be interesting the role of covid on invasive fungal infections ant the impact also in resistance. (see Segala FV, Bavaro DF, Di Gennaro F, et al. Impact of SARS-CoV-2 Epidemic on Antimicrobial Resistance: A Literature Review. Viruses. 2021;13(11):2110. Published 2021 Oct 20. doi:10.3390/v13112110)
- Conclusion: coherent with the results.
